# The Importance of the Kinematic Evaluation Methods of the Upper Limbs in Women with Breast Cancer Mastectomy: A Literature Review

**DOI:** 10.3390/healthcare11142064

**Published:** 2023-07-19

**Authors:** Israel Miguel-Andrés, María Raquel Huerta-Franco, Silvia Beatríz García-González, Miguel León-Rodríguez, Karla Barrera-Beltrán, Luis Angel Ortiz-Lango

**Affiliations:** 1Biomecánica, Centro de Innovación Aplicada en Tecnologías Competitivas, Leon C.P. 37545, Mexico; lortiz@ciatec.mx; 2Departamento de Ciencias Aplicadas al Trabajo, Universidad de Guanajuato, Campus Leon, Leon C.P. 37128, Mexico; mrhuertafranco@ugto.mx; 3Rehabilitación, Hospital Regional de Alta Especialidad del Bajío, Leon C.P. 37660, Mexico; beatrizgarglez@gmail.com (S.B.G.-G.); karla.barrera@hraeb.gob.mx (K.B.-B.); 4Departamento de Ingeniería en Robótica, Universidad Politécnica de Guanajuato, Cortazar C.P. 38496, Mexico; migueleon@upgto.edu.mx

**Keywords:** mastectomy, kinematics, physical treatment, breast cancer, rehabilitation

## Abstract

The kinematic assessment of the upper limbs in breast cancer (BC) survivors is one of the most common procedures to determine the recovery process after BC surgery. However, the methodology used is heterogeneous, finding various evaluation methods, which makes it difficult to compare results between studies. The objective of this review was to identify the technical features of the kinematic evaluation methods used in patients with mastectomy for BC. A literature review was conducted to search in electronic databases, such as PubMed, ScienceDirect, Clinical Key, Google Scholar, and Scopus. A total of 641 articles were obtained. After screening the title and the summary of the investigations, 20 manuscripts were kept for a deeper analysis. Different methodologies were found for the analysis of the kinematics of the upper limbs. Eight (40%) articles used the optoelectronic system, nine (45%) used the electromagnetic system, and three (15%) used other optoelectronic systems to assess shoulder kinematics. Each investigation studied different variables such as the type of surgery, the evaluation time, the age of the patients, the rehabilitation protocol, and so on. This makes the comparison among studies difficult, and the recovery process of the patients cannot be easily determined. In conclusion, the interpretation of the movement of the upper limbs should be easy to understand for oncologists, physiotherapists, clinicians, and researchers.

## 1. Introduction

Breast cancer (BC) is one of the most common types of cancer in women worldwide. Although there are several treatments to eradicate the tumor, mastectomy is one of the most chosen treatments by patients when they cannot take conserving treatments or when the treatments are not good enough. It has been shown that this surgical procedure affects the kinematics of the upper limbs (scapula, torso, and arm), which has repercussions on the quality of life of the women who suffer from this disease. Various studies have been performed to evaluate upper limb kinematics before or after mastectomy [1,2,3,4,5,6,7,8,9,10]. However, this information is scattered in the scientific literature. The kinematic evaluation of the upper limbs has been performed from weeks to years after mastectomy. In addition, the methodology used is heterogeneous, finding various evaluation methods, which makes it difficult to compare findings between studies. Although it is known that the mastectomy affects the kinematics of the shoulder, the most affected region of the upper limb has not been identified as some authors have analyzed the scapular motion [8], others the glenohumeral joint [9], and others the shoulder girdle [6]. Perhaps the type of surgery, the amount of lymph nodes extracted, or the muscle tissue affected is the main reason for the impairment of the kinematics of the shoulder.

Although several authors have reported results based on the three-dimensional electromagnetic tracking system [2,6,11], the comparison among different 3D tracking systems such as optoelectronic systems [1,4,8,9], 3D goniometers, or digital inclinometers [5] is difficult. This is because the description of the range of motion (ROM) of the upper limbs does not correspond to clinical angles (anatomical angles), making it difficult for physiotherapists and oncologists to comprehend the movement and implement specific physical treatments for BC patients. Moreover, several studies have commented that they follow the recommendations of the International Society of Biomechanics (ISB) for measuring the angles of the upper limbs. However, they do not describe the specific procedure to place the markers or electromagnetic systems, and define the coordinate systems [1,2,4,6,12]. The International Society of Biomechanics mentions that it is up to the researchers to relate the identification of the bony landmarks and equipment (optic or electromagnetic systems) used to measure the kinematics of the upper limbs [13]. Therefore, further information should be provided by the authors when they analyze the range of motion of the shoulder girdle before and after the mastectomy. The use of any equipment should specify how the bony landmarks are identified to create the coordinate systems.

It has been mentioned that physical therapy can help to recover the ROM of the upper limbs in patients after BC surgery [14,15]. However, specific physiotherapeutic treatments or protocols applied before or after breast cancer treatment have not been found. The analysis of the disability of the upper limbs should be easy for physiotherapists to understand. Therefore, physiotherapists can determine the specific physical treatment for the recovery process of patients who have undergone breast cancer surgery.

Therefore, the objective of this literature review is to identify the technical features of the kinematic evaluation methods used in patients with mastectomy for BC. The review was conducted with the Preferred Reporting Items for Systematic Reviews and Meta-Analyses (PRISMA) [16].

## 2. Materials and Methods

To analyze the technical specialized treatments and kinematic evaluation methods applied postmastectomy, a literature review was conducted searching electronic databases such as PubMed, ScienceDirect, Clinical Key, Google Scholar, and Scopus.

### 2.1. Inclusion Criteria

Observational and experimental studies, with or without a control group, were included in this review. The inclusion criteria considered manuscripts that evaluated the kinematics of the upper limbs of female participants who underwent mastectomy for breast cancer, with pre- and/or post-analysis. The studies considered in the analysis presented a kinematic evaluation with some motion capture systems.

After reviewing the articles, conference proceedings, reviews, duplicated manuscripts, books, theses, and studies written in languages other than English or Spanish were excluded from the analysis. Table 1 displays the inclusion criteria for the articles considered in the scientific literature review.

### 2.2. Search Strategies

A review of the scientific literature in English and Spanish was carried out using the information sources of PubMed, ScienceDirect, Clinical Key, Google Scholar, and Scopus. To identify published articles related to kinematic evaluations in patients with breast cancer, a review was conducted from 2005 to 2023, as shown in Table 2. The following search equation was used in English and Spanish: “kinematics” AND “breast cancer” AND “mastectomy”.

### 2.3. Selection Criteria for Scientific Manuscripts

During the search, specific keywords such as “kinematics”, “breast cancer”, or “mastectomy”, and “motion” or “three-dimensional movement” were required to appear in the title and abstract of the manuscripts. The abstracts were then evaluated to determine their relevance to the aim of this review for inclusion. To assess whether the studies met the selection criteria, two authors independently reviewed them and made decisions regarding inclusion or exclusion. Initially, based on the information provided in the title and summary of the manuscripts, a preliminary decision was made to include or exclude them. The authors of this literature review followed the guidelines outlined in the PRISMA guide during its creation [16,17].

### 2.4. Data Collection and Extraction

To identify the technical features of kinematic evaluations in patients with mastectomy for BC, significant information was extracted from the studies analyzed. The following variables were considered in the review: the objective of the study, the type of motion capture system (MoCS) used for the kinematic evaluation, the placement of markers and reference coordinate system considered in the studies, the time in which the kinematic assessment was performed (days, months or years before, or after mastectomy), the anthropometric characteristics of the population (age, sample size, type of surgery), the region of the body evaluated (shoulder, scapula, scapulothoracic joint), the movements evaluated, and the comparison of the kinematic results (healthy subjects, contralateral arm, premeasurement, and posttest).

## 3. Results

Based on the search performed in the five databases, a total of 641 articles were obtained, and 29 of them were duplicated studies. After screening the title and the summary of the investigations, 579 manuscripts were excluded, and five articles were not retrieved. In the end, 20 manuscripts were kept for a deeper analysis. Figure 1 shows the flow diagram of the literature review. Despite the rigorous methodology that was followed for the selection of the 20 scientific articles (see Figure 1), we observed that the objectives of these different studies were diverse, which we grouped into the following nine types:

(1) Studies where the objective was to evaluate the kinematics of the scapula in female BC survivors with mastectomy 6/20 (30%) [4,5,8,11,12,18]; (2) studies where the purpose was to assess shoulder kinematics 3/20 (15%) [1,2,9]; (3) investigations that aimed to assess the effect of lymphedema (in BC mastectomy survivors) on body posture and shoulder joint size [19], and to determine whether the presence of lymphedema decreased the range of shoulder motion 3/20 (15%) [6,20]; (4) one research study aimed to evaluate the effect of a home exercise program on upper limb function in survivors with mastectomy for BC [21]; (5) another study aimed to evaluate the effect of breast reconstruction on kinematics during functional tasks [7]; (6) the purpose of another research was to evaluate the kinematics between women with and without axillary web syndrome [22]; (7) other investigations aimed to evaluate the range of motion and strength follow BC treatment [23]; (8) one study had the objective of evaluating breathing movements of the thoracic and abdominal wall [24]; and (9) finally, another investigation whose purpose was to evaluate the muscular activity and the deviation of the shoulder movements in BC survivors with mastectomy [25].

In addition to the variety of research objectives in these 20 articles, we also observed that the vast majority of the investigations had different methodologies, which makes it difficult to make comparisons between the results; so, in this review article, we describe the following aspects: (1) the type of study and the methodological design (characteristics of the comparison groups), (2) the size of the study sample and the age of the participants, (3) the type of surgery performed and the time survivors of BC had since surgery at the time of the study, (4) the anatomical region that was evaluated, (5) the motion capture system used in the analysis, and (6) the kinematics of each evaluated joint (see Table 3).

### 3.1. Methodologies for Motion Capture System (MoCS), and Study Designs to Measure Kinematic Movements of the Upper Extremity in BC Survivors

By analyzing the methodologies described in the 20 manuscripts selected for this review article, we observed three types of techniques of MoCS for the evaluation of the kinematics of the upper extremity, in BC survivors with mastectomy, which are described below:

#### 3.1.1. The Optoelectronic System (Vicon Motion System)

In Table 3, we present the 8/20 (40%) studies that used the optoelectronic system technique to assess shoulder kinematics in mastectomy breast cancer survivors [1,4,7,8,9,12,18,23], and we describe the different studies below:

In 2019, the group of Lang et al. [1] evaluated the work-related functional task (overhead reach, repetitive reach, fingertip dexterity, hand and forearm dexterity, waist to overhead lift, overhead work), to assess three groups of participants (one group of BC survivors with impingement and another without impingement pain), and a control group of healthy subjects. In this investigation, the study groups were made up of 50 women with an average age of 52.8 ± 5.4 years (mean ± standard deviation of the mean); BC survivors had surgery 42.5 ± 41.6 months previously, and they underwent a mastectomy and lymph node removal. In this study, the anatomical regions evaluated were the torso (flexion/extension, lateral flexion/extension, axial rotation) and thoracohumeral joint (abduction/adduction, flexion/extension, internal/external rotation). The researchers used the recommendations of the International Society of Biomechanics (ISB) as a reference system and demonstrated that there were significant differences in the kinematics of the following groups: (a) the group of BC survivors who had pain and impingement; and (b) the two groups without pain. These investigators demonstrated that BC survivors who had impingement pain showed a significant reduction in overhead movement and upward rotation of the scapula (d = 0.80–1.11); they also observed that these patients showed a decrease in maximum humeral abduction and internal rotation movements in extreme postures (d = 0.54–0.78). Then, they concluded that impingement pain in BC survivors significantly influences the performance of functional shoulder tasks; therefore, it should be considered when evaluating pain as a potential factor for rotator cuff injuries in this type of patient.

In another study performed by Lang et al. in 2020 [12] using a methodology like that of their previous study [1], the authors evaluated three elevations in the frontal, scapular, and sagittal planes and the functional task (overhead reach, overhead lift, and fingertip dexterity) to assess a non-cancer control group and a BC survivor group (pain and no pain). In this investigation, the study groups were made up of 50 women with an average age of 53.1 ± 5.5 years; BC survivors had surgery 50.9 ± 45.7 months previously, and they underwent a mastectomy and lymph node removal. The anatomical regions evaluated were the scapula (upward rotation) and the scapulohumeral rhythm. The researchers used the ISB recommendations as a reference system and demonstrated that upward scapular rotation was reduced in patients with BC who had pain on forearm elevation levels in each plane up to 7.1° (*p* = 0.014 to 0.049); these authors described their results as inconsistent with the results of functional tasks, in which decreases in upward rotation were observed at higher levels of arm elevation; the investigators also demonstrated that angles of upward rotation and scapulohumeral rhythm during arm raising had a poor-to-moderate relationship (r = 0.003 to 0.970, *p* = 0.001 to 0.048) with functional task scores. The researchers also observed that arm elevation in the sagittal plane results in upward rotation of the scapula that was most closely associated with upward rotation during the performance of functional tasks. The authors concluded that these inconsistent relationships suggest that clinical assessments should adopt basic functional movements to assess the scapular movement to complement simple assessments of arm raises.

In 2022, Lang et al. [8] evaluated the functional movements (overhead reaching task) in three groups of subjects (control healthy group, mastectomy group, and breast reconstruction group). In their investigation, the study groups were made up of 95 women from 35 to 65 years of age; survivors with BC had undergone surgery >6 months prior; the anatomical regions that the investigators evaluated were the scapula (internal/external rotation, upward/downward rotation, and anterior/posterior tilt). The researchers used the ISB as the reference system and demonstrated that on the right side, the mastectomy-pain group had reduced upward rotation, while the reconstruction-pain group had higher upward rotation (mastectomy-only group: 22.9° vs. reconstruction: 31.2°; max difference = 8.3°, F = 13.1, d = 1.3, *p* < 0.001, compared to controls, the mastectomy-no pain, and the reconstruction-pain); on the left side, the mastectomy-pain group had higher internal rotation, while the reconstruction-pain group had reduced internal rotation (mastectomy-only: 45.1° vs. reconstruction group: 39.3°; max difference = 5.8°, F = 13.4, *p* = 0.01, d = 0.9); however, the time since surgery was longer in the mastectomy-pain group than the reconstruction-pain group, suggesting there may be a temporal component to kinematic compensations that on the right side, the scapular upward rotation was affected. The authors concluded that kinematic alterations in BC survivors may promote future rotator cuff disease development.

In 2022, García-Gonzalez et al. evaluated the flexion–extension and abduction–adduction movements of the glenohumeral (GH) joint in survivors of BC with mastectomy. In this investigation, the study group comprised 15 women with an average age of 46.7 ± 8.2 years; BC survivors underwent a mastectomy and axillary lymph node dissection and were evaluated 15 days before and 60 days after BC surgery. The investigators used three orthogonal coordinate systems (torso, left and right arm) as a reference. The researchers demonstrated that there was no significant difference in the range of motion of the GH joint when comparing pre-and post-mastectomy, flexion–extension (*p* = 0.138), and abduction–adduction (*p* = 0.058); however, patients who received chemotherapy (53%) before mastectomy were more affected (lower range of motion) than those who did not receive it. Therefore, the investigators concluded that the physical rehabilitation team must attend to these patients even before the mastectomy.

Brookham et al., 2018 [4] evaluated the tasks of shoulder ROM, daily life activity (DLA) tasks (personal body care activities), and work tasks (reaching tasks with and without loads) in two groups of BC survivors (affected side group and unaffected side group). In this investigation, the study groups were made up of 50 women with an average age of 59.4 ± 9.7 years; BC survivors had completed treatment > 3 months prior; and they underwent mastectomy, lumpectomy, and axillary node dissection. The authors evaluated the following anatomical regions: (i) the scapulothoracic joint (upward/downward rotation, anterior/posterior tilt, retraction/protraction) and (ii) the humerothoracic angle (elevation angle; the plane of elevation: flexion, abduction; external/internal rotation). The investigators used the ISB recommendations as a reference system and demonstrated that the affected side had reduced ROM in the plane of elevation (32.3° vs. 39.0°, *p* = 0.003) in ROM-reach tasks, as well as in elevation angle and the plane of elevation in ROM-rotate tasks (9.7° vs. 12.0°, *p* = 0.0121; and 15.3° vs. 18.6°, *p* = 0.04) in maximal humerothoracic angles. The affected side had reduced angles of elevation (48.4° vs. 54.9°, *p* < 0.0001), as well as less external rotation during work tasks (0.4° vs. 9.3°, *p* = 0.008); when the authors evaluated the scapulothoracic kinematics, differences were observed between the affected and unaffected scapulothoracic ROM; the affected side had increased anterior/posterior tilt ROM in DLA and work tasks (16.2° vs. 14.4°, *p* = 0.043; and 16.6° vs. 14.6°, *p* = 0.03, respectively). The researchers concluded that a reduced range of motion on the affected side suggests the BC population had less varied movement strategies, keeping movements in narrower ranges to avoid disability, pain, or subacromial impingement.

Maciukiewicz et al., 2022 [23], evaluated arm flexion, extension, abduction, adduction, scapular abduction, and internal external rotation in two groups of BC survivors (one year post-treatment survivors BC group, and between one and two years since treatment ended group); the study groups were made up of 29 women with an average age of 53.9 ± 10.3 years; the BC survivors had surgery >1 year prior and had undergone any form of surgical procedure for breast tumor removal. These investigators evaluated the humerothoracic joint range of motion (flexion, extension, scapular abduction, abduction, internal and external rotation) and used the ISB recommendations as a reference system to demonstrate that the time since the treatment influenced the strength when performing the movements of the humerothoracic joint. The researchers also showed that the range of movement (during flexion and scapular abduction) in the group that had more treatment time resulted in 11.5–15.5° with less range of movement and 27.7–43.6 N less force production. Therefore, the authors proved that the effect of the time elapsed since the treatment influenced the muscular performance during the tasks being carried out, and they proved that the activation was much greater in the group that had undergone treatment 1 to 2 years prior. For this reason, the investigators concluded that the effects of treatment can manifest late, so the strength and range of movement are reduced in BC survivors, especially in those who have stopped treatment for more than 1 year.

Another study by Lang et al., 2022 [7], evaluated the maximum arm abduction and extension and functional tasks (right and left repetitive reach, right and left fingertip dexterity, overhead lift, and overhead work) in three groups of participants (BC survivors with mastectomy, BC patients with mastectomy and breast reconstruction, and a control group); the study groups were made up of 95 women with an average age of 54.0 ± 5.1 years. These investigators demonstrated that BC survivors had undergone surgeries 56.8 ± 42.1 months prior, which were mastectomy and lymph node removal, and one group was post-mastectomy reconstruction. The anatomical regions evaluated were the shoulder range of motion (abduction and extension); the thoracohumeral joint (abduction, horizontal flexion, axial rotation); and scapular motion (protraction/retraction, upward/downward rotation). Likewise, the investigators used the ISB recommendations as a reference system and demonstrated that both BC groups had higher self-reported disability than the controls did (*p* < 0.001), and the arm extension range of motion was lower in both BC groups (*p* < 0.001; d = 0.88). The reconstruction group had the lowest performance outcomes during the repetitive reach (*p* = 0.002; Cohen d = 0.94), overhead lift (*p* = 0.038; d = 0.83), and overhead work (*p* < 0.001; d = 1.1). The abduction range of motion was reduced only in the reconstruction group (*p* < 0.001; d = 0.63). The investigators concluded that patients with BC display kinematic alterations after reconstructive surgery that may contribute to functional performance decreases and create instability in the shoulder, so they suggested that postoperative rehabilitation should focus on restoring function to the shoulder by addressing the imbalances created by surgery.

#### 3.1.2. Electromagnetic System to Evaluate Kinematic Movements of the Upper Extremity in BC Survivors

Table 3 also presents the 9/20 (45%) investigations in which the electromagnetic system technique was used to evaluate the kinematics movements of the upper limbs in BC survivors with mastectomy [2,5,6,11,20,22,25,26,27]; below, we present the details of the methodological design used in these investigations:

Crosbie et al., 2010 [2], used the motion star wireless (electromagnetic system) technique to evaluate forward flexion, abduction, and movement in the scapular plane of three groups of participants (a healthy control group, a group with dominant side mastectomy, and a group with nondominant side mastectomy); the study groups were made up of 75 women with a range age of 44–88 years. The authors demonstrated that BC survivors had undergone surgery >12 months prior; all participants had undergone a mastectomy, post-mastectomy reconstruction, and lymph node removal. In this investigation, the anatomical regions evaluated were scapulothoracic (upward/downward rotation, internal/external rotation, anterior/posterior tilt) and trunk (flexion, extension, lateral flexion, axial rotation of thoracic and lumbar region). The authors demonstrated that the survivors of BC mastectomy groups had significant differences in upward rotation on both sides when compared to the control group (CtlG); on the dominant side, both mastectomy groups had greater up/down rotation compared to the CtlG, but the two mastectomy groups were not different from each other; postmastectomy women showed altered patterns of scapular rotation compared with CtlG in all planes of motion; in particular, the scapula on the mastectomy side rotated upwards to a markedly greater extent than on the non-mastectomy side, and post-mastectomy women exhibited greater excursion than the controls. These investigators concluded that the impaired movement pattern is related to mastectomy on the same affected side. For this reason, the authors suggest carrying out more research in which rehabilitation interventions designed to restore normal scapulohumeral relations were applied, especially on the affected side, after unilateral mastectomy for BC.

Shamley et al., 2009 [11], used the technique of the Polhemus Fastrak (electromagnetic system) to evaluate the humeral elevation and depression in the scapular plane in two groups of BC survivors with mastectomy (one group: affected side, left and right, and other groups: unaffected side, left and right); the study groups were made up of 152 women with an average age of 61.8 ± 8.9 years; BC survivors had undergone surgery 1144.0 ± 537 days prior; all participants had undergone a mastectomy and wide local excision. The authors evaluated the scapula (protraction/retraction, lateral/medial rotation, anterior/posterior tilt) and used the ISB protocol (orientation of the scapula relative to the trunk) as a reference system and demonstrated that movement patterns of protraction/retraction, and the rotation of the scapula showed opposite effects when the right side was affected. The researchers also proved that all scapulothoracic movements were significantly altered on the affected side compared to the unaffected side and were independent of the type of medical treatment; the difference was significantly greater when the left side was affected. These authors proved that left scapulothoracic dysfunction included increased protraction, increased posterior tilt, and decreased lateral rotation, these results being significantly associated with having received chemotherapy. These investigators also proved that right scapulothoracic dysfunction included increased retraction, increased posterior tilt, and increased lateral rotation; differences in scapulothoracic lateral rotation were associated with downward movement. In this study, both pain and disability were proven to be associated with scapulothoracic dysfunction. The researchers concluded that patients treated for breast cancer showed shoulder pain and movement dysfunction and that patients with left-sided carcinoma are the group most likely to develop more pain and dysfunction after treatment; so, the authors concluded that whether cause or effect, pain reports are accompanied by three-dimensional scapula dysfunction which mimics that of many other shoulder conditions.

Another study by Shamley et al., made in 2012 [26], used the three Space Fastrak (electromagnetic system) technique to evaluate the elevation and depression arm and scapular plane in two groups of participants (survivors of BC, mastectomy group, and a group with wide local excision; both left and right affected sides); the study groups were made up of 176 women with an average age of 61.6 ± 9.1 years; BC survivors had undergone surgery 1143.81 ± 534.77 days prior. These investigators evaluated the scapula (internal/external rotation, upward/downward rotation, and anterior/posterior tilt) and used the ISB protocol standard (thorax, scapula, humerus) as a reference system. The authors demonstrated that patients had greater upward rotation on the affected sides compared to healthy participants (CI 4.82–8.51 for the left hand; CI 3.91–7.70 for the right hand, *p* < 0.0001); the investigators demonstrated that unaffected shoulders of BC survivors also showed greater upward rotation (left hand, CI 4.61–8.23, *p* < 0.001; right hand, CI 3.05–6.92, *p* < 0.001) and decreased posterior tilt (left hand, CI −0.40–1.15, *p* < 0.34; right hand, CI 0.98–2.97, *p* < 0.001) than the healthy shoulders; thus, both shoulders in the patients had movement deviations over and above the normal variation; the authors suggest that differences between the tilt of the affected and unaffected shoulders of the patients were significantly associated with pain and disability. These investigators concluded that shoulder morbidity is bilateral, greater in patients with a mastectomy, and is present for up to six years post-surgery; this study demonstrated evidence to support prospective surveillance programs that can be integrated into survivorship programs.

In 2021, Baran et al., at the University of Ankara, Turkey [6], used the flock of birds technique (electromagnetic system) to assess the elevation in the scapular plane of three groups of BC survivors with a mastectomy: (i) a group without lymphedema, (ii) a group with moderate lymphedema, and (iii) a group with severe lymphedema. In this investigation, the study groups were made up of 67 women with an average age of 50.9 ± 7.3 years; in this study, BC survivors had undergone surgery 32 ± 26.8 months prior; all participants were modified radical or radical mastectomy and axillary lymph node dissection; the anatomical regions evaluated were the scapula (internal–external rotation, upward–downward rotation, and anterior–posterior tilt). These researchers used the ISB as a reference system and demonstrated that the scapular upward rotation was less for the severe lymphedema group than the non-lymphedema group in the 90–60–30° depression phases of arm elevation (*p* < 0.05). The authors also observed that the anterior tilt of the scapula was greater for the severe lymphedema group than for the non-lymphedema group at the 30° dip phase of arm raising (*p* < 0.05). The shoulder abduction range of motion was lowest in the severe lymphedema group (*p* < 0.05). The investigators demonstrated that the group without lymphedema had the lowest Quick-DASH score and the group with severe lymphedema had the highest score (*p* < 0.05). The researchers also proved that there was a moderate association between Quick-DASH pressures and scapular movements in all groups studied (*p* < 0.05). This investigation proved that the development, presence, and/or severity of lymphedema is associated with alterations in the kinematics of the shoulder girdle and a decrease in the function of the upper extremities. Therefore, it is advisable to implement longitudinal studies, in which the following factors are controlled: (i) the presence of lymphedema, (ii) the time after mastectomy, (iii) the dominant limb, and (iv) the presence of shoulder impingement [1].

In 2022, Braudy et al. [22] used the Polhemus three Space FASTRAK (electromagnetic system) technique to evaluate arm forward flexion, scapular plane abduction, and coronal plane abduction in two groups of BC survivors with mastectomy: a group of axillary web syndrome (AWS) and a non-AWS group; the study groups were made up of 25 women with an average age of 54.0 ± 10 years. The researchers demonstrated that the BC survivors had undergone surgery >5 years prior; all participants had undergone lumpectomy, mastectomy, and axillary surgery, and the anatomical regions evaluated were scapulothoracic (internal and upward rotation and posterior tilt) and humerothoracic (elevation, elevation plane, and axial rotation). The researchers used the ISB recommendations as a reference system and demonstrated that women with AWS had 15.2° less scapular upward rotation at a 120° humerothoracic elevation (95% confidence interval [−25.2–5.2], *p* = 0.005), regardless of the plane. No significant differences were observed between the groups for any other angle of scapular upward rotation, nor scapular internal rotation, scapular posterior tilt, or glenohumeral axial rotation at any angle. The researchers concluded that 5 years after surgery for BC, women diagnosed with AWS have altered scapulohumeral kinematics that may place the mat at an increased risk of shoulder pain based on the existing kinematic literature in healthy cohorts; therefore, the authors suggest that the results of their research can help guide rehabilitation programs for BC survivors to facilitate pain-free upper extremity function after treatment.

In 2019, Ribeiro et al. [5], used the flock of birds technique (electromagnetic system), to evaluate the elevation of the arm in the scapular plane in two groups of BC survivors, postmastectomy (a control group, and a surgery group); the study groups were made up of 42 women with an average age of 50.2 ± 9.8 years; BC survivors had undergone surgery 24.2 ± 20.5 months prior; all participants were conserving surgery or mastectomy. In this investigation, the anatomical region evaluated was the scapula (internal/external rotation, upward rotation, anterior/posterior tilt). The investigators used the recommendations of the ISB as a reference system and demonstrated that for scapular upward rotation, there was an interaction between the groups vs. the angle during the elevation (*p* = 0.02, F = 2.57) of the arm. The authors also observed that the affected side of the surgery group showed less upward rotation compared to the control group at 120° of arm elevation (*p* = 0.01, mean difference = 7.3°, Cohen’s d = 0.47). The authors also demonstrated that the surgery group decreased the range of external rotation in the affected limb when compared to both the non-affected limb (*p* = 0.001) and the controls (*p* = 0.01). The authors demonstrated that patients in the surgery group had decreased muscle strength in shoulder abduction movements in the affected limb compared to the unaffected limb (*p* = 0.03) and observed that the affected side of the surgical group had decreased external rotation muscle strength compared to the unaffected limb (*p* = 0.01). The authors of this study concluded that upward scapular rotation decreases by 120°; they also demonstrated that shoulder external rotation, abduction force, external rotation force, function, and quality of life are affected in these women. Therefore, their results prove that the scapular kinematics is altered during the elevation of the arm in the scapular plane, and there is a restriction of the range of movement and muscular strength deficit in these patients.

In 2016, Spinelli et al., from the University of Rhode Island Hospital, USA [27], used the Liberty Polhemus technique (electromagnetic system) to evaluate the functional task (unweighted and weighted overhead reaching, and simulated hair combing) in a group of BC survivors with mastectomy compared with a control group. The study groups were made up of 60 women with an average age of 53.8 ± 10.9 years; the BC survivors had undergone surgery 29.4 ± 10.8 mean months prior; all participants had undergone lumpectomy, mastectomy, and lymph node surgery. The investigators evaluated the following anatomical regions: scapulothoracic (ST) range of motion (elevation, internal/external rotation, clavicular elevation/retraction, upward rotation, posterior tilt) and glenohumeral (GH) range of motion (adduction, external rotation). The investigators demonstrated no significant differences in ranges of motion of the scapulothoracic or glenohumeral joints between women with and without a history of BC during unweighted (*p* = 0.32) and weighted (*p* = 0.51) reaching movements, and the task of combing their hair (*p* = 0.76). The authors proved that on average, the scapula rotated upward and tilted backward while the clavicle rose and retracted and observed that minimal scapular internal/external rotation occurred. The authors proved that in the motion of the glenohumeral joint, the humerus was elevated, abducted, and externally rotated during the functional tasks performed in this study. They also demonstrated a significant correlation between pain subscale scores and the range of motion of upward rotation of the scapulothoracic joint during unweighted reaching (*p* < 0.05) and the range of motion of clavicular retraction during the task of combing their hair (*p* < 0.05). The pain subscale scores were correlated with the glenohumeral joint external rotation range of motion during unweighted reaching movement (*p* < 0.05), weight reaching (*p* < 0.05), and the task of combing their hair (*p* < 0.05). The functional task subscale scores were correlated with glenohumeral joint external rotation during unweighted (*p* < 0.05) and weighted (*p* < 0.05) reaching.

Rundquist et al., 2015 [20], used the motion monitor 3D (electromagnetic system) technique to assess shoulder flexion, abduction, and external and internal rotation in BC survivors with mastectomy (a group that involved upper limbs, and another group with uninvolved upper extremities). In this investigation, the study groups comprised 30 women with an average age of 57.8 ± 10.1 years; BC survivors had undergone surgery 72.3 ± 64.3 months prior; the anatomical region evaluated was the shoulder range of motion (flexion, abduction, external and internal rotation). These researchers do not mention what they used as a reference system; however, they demonstrated significant differences in the external rotation (ER) of the involved shoulder 53.69 ± (20.97) vs. the non-involved shoulder 64.89 (20.59), t = −2.34, *p* = 0.03; they also observed that there was a significant difference in ER (*p* = 0.026) and volume (*p* < 0.001) between involved and uninvolved upper extremities; the mean ER was 11.2° less in the involved side. In this study, the mean difference in arm volume was 368.8 mL greater on the involved side. In addition, the authors describe that this study has some limitations related to a small sample size, and they concluded that there were significant differences in ER between the involved and uninvolved arms; therefore, the authors recommend further research with a more significant number of participants to identify functional differences in this population.

Another study by Shamley et al., 2014 [25], used the three Space Fastrak (electromagnetic system) technique to evaluate arm elevation and depression in the scapular plane in two groups of women surviving BC with mastectomy (affected sides, left and right, and another group, unaffected sides, left and right); the study groups were made up of 176 women with an average age of 61.6 ± 9.1 years. In this study, the BC survivors had undergone surgery 1143 ± 534.7 days prior; all participants had undergone a mastectomy and wide local excision. These investigators evaluated the scapula (internal/external rotation, upward/downward rotation, anterior/posterior tilt). The authors used the International Shoulder Group (ISG) protocol as the reference system and demonstrated that when the right side is affected, the scapula has a greater externally rotated and anteriorly tilted starting position and remains more externally rotated throughout the movement; so, the investigators suggest that the movement into posterior tilt is delayed over the first 50° of elevation. They consider that this movement pattern is accompanied by the following factors: (i) reduced muscle activity in the upper trapezius (UT), (ii) increased activity in the pectoralis major (PM), and (iii) the earlier release of serratus anterior (SA) activity. When the authors compared the left affected side versus the right unaffected side, they demonstrated that during elevation, the left affected side lost approximately 10° of external rotation and showed a reduced range of posterior tilt during the critical phase of elevation (80°–120°) as well as lowering the arm. The investigators also proved that having received chemotherapy contributes significantly to the difference seen between the affected and unaffected shoulders in patients. In this study, differences between the tilt of affected and unaffected shoulders in patients were also demonstrated to be significantly associated with pain, disability, and changes in SA activity. When the authors compared healthy shoulders vs. mastectomy patients, they demonstrated increased activity in both the left and right affected shoulders in all muscles (PM, CI [8.77–3.66], *p* < 0.001; UT, CI [22.97–14.30], *p* < 0.001; rhomboids muscles, IC [15.38–12.07], *p* < 0.001; SA, IC [10.36–5.48], *p* < 0.001), whereas, in the case of wide local excision (WLE), those increases were not observed in the SA and PM activity on the right affected shoulders, where a decrease was noted.

#### 3.1.3. Other Optoelectronic Systems to Measure the Kinematics Movements of the Upper Extremity in BC Survivors

In 2018, Corrado et al. [21] used the SMART-DX (optoelectronic system) to evaluate functional tasks (hand-to-mouth, reaching-arm, hand-to-head) and ROM tasks (shoulder flexion/extension, abduction/adduction, elbow flexion/extension) in a group of BC survivors with mastectomy (one group without a home exercise program, and another group with home exercise program). The study groups were made up of 30 women of 55.8 mean years of age. The BC survivors had undergone surgery 1–3 months prior. All participants had undergone modified radical mastectomy and axillary lymph node dissection. In this study, the anatomical regions evaluated were the shoulder and the elbow (movement duration and angular velocity); the investigators used the ISB system as a reference and demonstrated that in the hand-to-mouth test, the duration of the movements (mean ± SD) was faster in the group that exercised at home (0.76 ± 0.22 s) compared to those that did not (1.26 ± 0.06, *p* = 0.02). The evaluation at three months showed that the duration of this test in seconds was 0.71 ± 0.24 s vs. 1.62 ± 0.37 s for the groups of exercise and no-exercise, respectively; *p* = 0.023. The results of the reaching-arm test showed similar results after a month of exercising, and the times were 0.86 ± 0.28 s vs. 1.75 ± 0.28 s; *p* = 0.02. At three months, the results were 0.70 ± 0.13 s vs. 1.57 ± 0.3, *p* = 0.01 for the groups that practiced exercise and no exercise, respectively. In the hand-to-head, one month after performing the exercises, the results were 0.67 ± 0.37 vs. 1.88 ± 0.1, *p* = 0.005. After three months of exercise, the results were 0.55 ± 0.35 vs. 2.06 ± 0.24, *p* = 0.003 for the group that exercised and did not exercise, respectively. The researchers concluded using three-dimensional motion analysis that the home exercise program is an effective tool for preventing upper extremity dysfunction in the breast.

Balzarini et al. [19] used the technique of ELITE 2002 (optoelectronic system), but they do not clearly describe the movements of the upper extremity that was evaluated. The study groups were made up of 17 women with an average age of 58.9 mean years; all participants underwent quadrantectomy and modified radical mastectomy. The investigators do not mention the anatomical regions that were evaluated, nor the system reference used. However, their results demonstrated a limited range of motion of the affected arm, a reduction in swinging during walking tests, and in shoulder retroposition and abduction movements for all patients; also, it was shown that in the retroposition test that the angle obtained for the healthy shoulders was an average of 54.6 ± 10.9° vs. 46.0 ± 9.91° for the shoulders on the surgical side, *p* = 0.009. These authors demonstrated that after repeated cyclical movements, premature fatigue appeared in the pathological arm. Lymphedema does not appear to cause alterations to the posture of the spine in the participants, but the drooping of the shoulder homolateral to the lymphedema can occur. The researchers concluded that this kind of investigation is quick, easy, and comfortable for patients with lymphedema and can be a useful method to evaluate functional capacity, thus allowing a quantitative assessment of the loss of function and the optimization of the rehabilitative protocol.

Lopot et al., 2019 [24], used the Qualisys system (optoelectronic kinematic analyzer), to evaluate the mild and deep breath movements of the thoracic and abdominal wall in two groups of participants (control healthy group and a group of survivors of the BC with mastectomy); the study groups were made up of 12 women with an average age of 64.5 years. In this study, the BC survivors had undergone surgery >5 years prior; all participants were submitted to total breast mastectomy, and the anatomical regions evaluated were the thoracic and abdominal walls. The researchers do not mention which reference system they used; however, their results demonstrated that mastectomy affects the extent of breathing movements in women survivors of BC with mastectomy by reducing the range of breathing movements on the side of the surgery. They also demonstrated that the symmetry of the range of movement (ROM) between the surgery and non-surgery sides during breathing is also impaired. They point out that the most striking asymmetries are present approximately on the level of the fifth ribs, where the postoperative scar is the most common. The authors describe that these findings have been confirmed in both quiet and deep breathing, demonstrating their claim to the need for post-surgical scar care of female BC survivors.

## 4. Discussion

It has been shown that normal movements of the arm and shoulder require normal mobility of the following joints: (a) the scapulothoracic (ST), (b) the glenohumeral (GH), (c) the acromioclavicular (AC), and (d) the sternoclavicular (SC) joints [11]. In upper extremity kinematics, the humerus moves synchronously and is compatible with the scapula [6]. The shoulder mechanism involves a combination of rotations and translations about these four regions resulting in three-dimensional movements. Under healthy conditions, raising the arm is accompanied by the retraction of the scapula, lateral rotation, and posterior tilt; however, when scapulothoracic movement is disproportionate to glenohumeral movement, there is a potential risk for microtrauma and long-term pain [11]. Baran et al. describe that in 1996, Inman and Abbott reported that the elevation of the glenohumeral joint and the scapulothoracic rotation is 2:1; the ratios between normal scapulohumeral joints and damaged shoulder joints ranged from 1.35:1 to 7.9:1 [6].

Alterations in the shoulder joint in mastectomy BC survivors is a well-proven fact [1]. Studies show that women who have undergone a mastectomy have decreased scapular upward rotation, external rotation, and posterior tilt in the scapular plane during humerothoracic lift; therefore, the scapulohumeral rhythm is disturbed [1,2,6,11]. When the scapulothoracic movement is disproportionate to the glenohumeral movement, there is the possibility of microtrauma, chronic pain [10], and rotator cuff disease (RCD) [1]. A series of studies in women treated for BC show limited glenohumeral range of motion. In 2009, Shamley et al. demonstrated decreased activity in the following four muscles for scapular movements: (1) the serratus anterior muscle, (2) the upper trapezius (UT), (3) the pectoralis major, and (4) the rhomboid muscles. The researchers found a marked reduction in UT and rhomboid activity, followed by shoulder pain and disability; they demonstrated that the pectoralis major and minor muscles were atrophied on the side affected by cancer, findings that suggest alterations of the biomechanics of the shoulder complex [11].

These kinematic impairments depend on certain factors such as the type of surgery performed on the patients. According to Shamley et al. [26], patients who underwent a mastectomy vs. a wide local excision (WLE) presented a greater deviation of movement and pain. Moreover, Lang et al. [7,8] compared scapular kinematics in patients with mastectomy vs. mastectomy plus reconstruction, finding that patients with reconstruction presented more kinematic alterations. From the articles included in this review, only these two make a specific comparison between groups of different surgical interventions, even though several of the articles included in this review included different types of surgeries performed [7,8]. The type of surgical procedure can affect the kinematics of the shoulder girdle. Although the treatments focus mainly on the breast tissue, they can affect the fascia, pectoral muscle tissue, ligaments, tendons, or nerves [22]. Therefore, it can be assumed that the more complex the surgical procedure, the greater the affectation on the kinematics, and not only considering surgery but also local approaches, such as radiation can produce fibrosis in the tissues, affecting their mobility. Furthermore, chemotherapy could produce weakness and fatigue, as well as peripheral neuropathy [9]. Another factor to consider is the presence of lymphedema after mastectomy and axillary lymph node dissection (ALND), where, according to Baran [6], the effects on scapular kinematics increase with the severity of the lymphedema.

According to Lang et al., 2022 [18], body mass index (BMI) can influence the monitoring of the scapular movement. The presence of adipose tissue makes it difficult to locate and palpate the bony landmarks for the placement of markers. Therefore, people with overweight or obesity may present greater limitations during the kinematic evaluation. Finally, we must not forget that the diagnosis and treatment of patients with breast cancer have an emotional and social impact, caused by the affectation of their body image. These factors can produce a negative effect on the kinematic movements of the upper limbs [9].

## 5. Conclusions

After reviewing the manuscripts, it is evident that several methodologies and pieces of equipment can be applied for the analysis of the kinematics of the upper limbs. Every piece of equipment or system uses these techniques, physical principles, and reference coordinate systems to track the movement of the limbs. Furthermore, each study presented different variables such as the type of surgery, the evaluation time after or before surgery, the age of the patients, rehabilitation protocol in some cases, daily life tasks, comorbidities, the specific region of the shoulder analyzed, and so on. These different variables make it difficult to compare among the studies, and the recovery processes of the patients cannot be easily determined. Moreover, one of the limitations present in most of the manuscripts is the limited number of patients with mastectomy involved in each study.

The range of motion of the shoulder girdle (scapula, acromioclavicular joint, glenohumeral joint, clavicle, torso, sternoclavicular joint) is one of the common variables used to measure the recovery process of patients with breast cancer surgery. However, the different equipment employed to measure the movements of the upper limbs hinders the comparison among studies. Furthermore, although several studies consider the recommendations of the International Society of Biomechanics (ISB) for the interpretation of the movement of the upper limbs, the technical procedure of the methodology for the coordinate systems is still complex and difficult to understand. Most of the authors do not describe the technical procedure in detail for an easy understanding of the relative motion of the upper limbs.

In conclusion, it is important that the interpretation of upper limb movement be easily understandable for medical staff. This understanding would enable them to determine the specific physical treatment to apply for the recovery process of patients who have undergone breast cancer surgery. Furthermore, the establishment of a standardized methodology would facilitate the comparison of upper limb kinematics after mastectomy.

## Figures and Tables

**Figure 1 healthcare-11-02064-f001:**
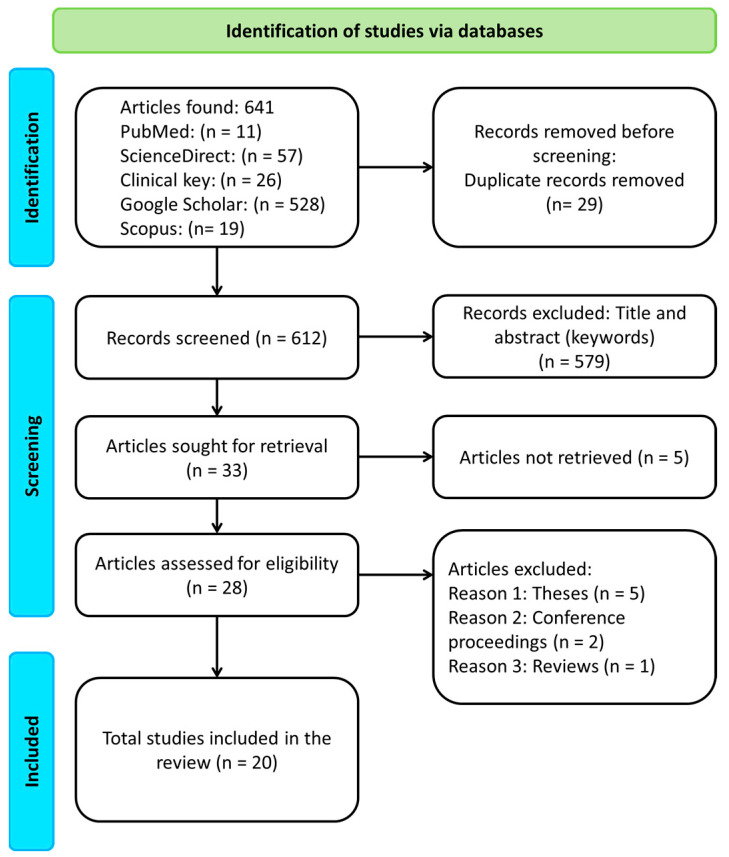
Flow diagram for the literature review, PRISMA 2020 [17].

**Table 1 healthcare-11-02064-t001:** Inclusion criteria were used to include articles in the review.

**Study design**	Clinical studies, and observational trials (cross-sectional or longitudinal)
**Target population**	Women with mastectomy for breast cancer
**Treatment**	Mastectomy
**Variables analyzed**	Kinematics of the glenohumeral joint, scapula, or shoulder girdle
**Comparison**	Control group (healthy women, contralateral side). Pre and post-test measurement

**Table 2 healthcare-11-02064-t002:** Search strategy in different databases.

Database	Search Equation	Search Date	Outcomes	Selected
**PubMed**	“kinematics” AND “breast cancer” AND “mastectomy”	4 March 2023	11	9
**ScienceDirect**	“kinematics” AND “breast cancer” AND “mastectomy”	2 March 2023	57	8
**Clinical Key**	“kinematics” AND “breast cancer” AND “mastectomy”	3 March 2023	26	8
**Google Scholar**	“kinematics” AND “breast cancer” AND “mastectomy”	17 February 2023	528	27
**Scopus**	“kinematics” AND “breast cancer” AND “mastectomy”	5 March 2023	19	15
**Total**			641	67

**Table 3 healthcare-11-02064-t003:** Articles considered in the literature review.

Articles	Objective	MoCS and Kinematics (Movements)	Reference System	Time (Days or Months)	Age and Sample Size	Surgery	Region of the Body	Comparison
Lang et al., 2022 [8]	Scapular kinematics and impingement pain during an overhead reach task	Vicon motion System (optoelectronic system) Functional movement (overhead reaching task)	Recommendations of the International Society of Biomechanics (ISB) (torso cluster)	>6 months after surgery	35–65 years old *n* = 95	Mastectomy and reconstruction post-mastectomy	Scapula (internal/external rotation, upward/downward rotation and anterior/posterior tilt)	Control healthy group; Mastectomy group; Reconstruction group
Crosbie et al., 2010 [2]	Shoulder girdle kinematics and control group	Motion star wireless (electromagnetic system) Forward flexion, abduction, movement in the scapular plane	Local coordinate system, 6-degree-of-freedom	>12 months after surgery	44–88 years old n = 75	Mastectomy	Scapulothoracic (upward/downward rotation, internal/external rotation, anterior/posterior tilt). Trunk (flexion, extension, lateral flexion, axial rotation of thoracic and lumbar region)	Control healthy group; Mastectomy dominant side; Mastectomy nondominant side
Shamley et al., 2012 [26]	Impact of a mastectomy vs. a wide local excision	3 Space Fastrak (electromagnetic system) Elevation and depression arm in the scapular plane	ISB standard (thorax, scapula, humerus)	1143.81 (534.77) days after surgery	61.6 (9.1) years old n = 176	Mastectomy and wide local excision	Scapula (internal/external rotation, upward/downward rotation and anterior/posterior tilt)	Mastectomy group; Wide local excision group (Both left and right affected sides)
García-González et al., 2022 [9]	Shoulder kinematics before and after mastectomy	Vicon Nexus System (optoelectronic system) Flexion-extension and abduction-adduction movement	Three orthogonal coordinate axes systems (torso, left and right arms)	<15 days before surgery 60 days after surgery	46.7 (8.2) years old n = 15	Mastectomy and axillary lymph node dissection	Glenohumeral joint (flexion/extension and abduction/adduction)	Mastectomy group before and after surgery
Corrado et al., 2018 [21]	Effects of home exercise program on upper limb function	SMART-DX (optoelectronic system) Functional task (hand-to-mouth, reaching-arm, hand-to-head) ROM task (shoulder flexion/extension, abduction/adduction, elbow flexión/extension)	International shoulder groups (ISB) protocol	1 and 3 months after surgery	55.8 mean age n = 30	Modified radical mastectomy and axillary lymph node dissection	Shoulder and elbow (movement duration and angular velocity)	Patients without home exercise program; Patients with home exercise program
Shamley et al., 2009 [11]	Scapulothoracic kinematics between affected and unaffected side	The Polhemus Fastrak (electromagnetic system) Humeral elevation and depression in the scapular plane	ISB protocol (orientation of the scapula relative to the trunk)	1144 (537) mean days after surgery	61.8 (8.9) mean years n = 152	Mastectomy and wide local excision	Scapula (protaction/retraction, lateral/medial rotation, anterior/posterior tilt)	Affected side left and right; Unaffected side left and right
Balzarini et al., 2006 [19]	Alterations in postural strategies due to increased weight and volume of the arm due to lymphedema	ELITE 2002 (optoelectronic system) Shoulder retroposition and abduction movements	Global coordinate system, trajectories (X, Y, Z) of the markers	The evaluation time after the mastectomy is not clear.	58.9 mean age n = 17	Quadrantectomy and modified radical mastectomy	ROM of the shoulder girdle and the affected arm.	Unaffected side in the mastectomy group
Baran et al., 2021 [6]	Breast cancer-related lymphedema on shoulder girdle kinematics	Flock of birds (Electromagnetic system) Elevation in the scapular plane	Recommendations of the International Society of Biomechanics (ISB)	32.5 (26.8) mean months since surgery	50.9 (7.3) mean age n = 67	Modified radical or radical mastectomy and axillary lymph node dissection	Scapula (internal-external rotation, upward-downward rotation and anterior-posterior tilt)	Non lymphedema group; Moderate lymphedema group; Severe lymphedema group
Braudy et al., 2022 [22]	Kinematics between women with and without axillary web syndrome	Polhemus 3Space FASTRAK (electromagnetic system) Arm forward flexion, scapular plane abduction, coronal plane abduction	ISB recommendations	>5 years after surgery	54 (10) Mean age at surgery n = 25	Lumpectomy, mastectomy, and axillary surgery	Scapulothoracic (internal and upward rotation and posterior tilt) Humerothoracic (elevation, elevation plane, and axial rotation)	Axillary web syndrome; Non-axillary web syndrome
Lang et al., 2020 [12]	Scapular upward rotation and scapulohumeral rhythm during arm elevation	Vicon Motion System (Optoelectronic system) Three elevations in frontal, scapular, and sagittal plane. Functional task (overhead reach, overhead lift, and fingertip dexterity)	Recommendations of the International Society of Biomechanics (ISB)	50.9 (45.7) mean months since surgery	53.1 (5.5) mean age n = 50	Mastectomy and lymph node removal	Scapula (upward rotation) and Scapulohumeral rhythm	Non-cancer controls group; Breast cancer survivors group (pain and no pain)
Lang et al., 2019 [1]	Torso and shoulder kinematics during common task	Vicon Motion System (Optoelectronic system) Work-related functional task (overhead reach, repetitive reach, fingertip dexterity, hand and forearm dexterity, waist to overhead lift, overhead work)	Recommendations of the International Society of Biomechanics (ISB)	42.5 (41.6) months since surgery	52.8 (5.4) mean age n = 50	Mastectomy and lymph node removal	Torso (flexion/extension, lateral flexion/extension, axial rotation) Thoracohumeral (abduction/adduction, flexion/extension, internal/external rotation) Scapula (protraction/retraction, upward/downward rotation, anterior/posterior tilt)	Breast cancer survivors (with and without impingement pain) group; Control group
Lang et al., 2022 [18]	Accuracy of the AMC (acromion marker cluster) for scapula motion tracking	Vicon Motion System (Optoelectronic system) Humeral elevation in frontal plane	Recommendations of the International Society of Biomechanics (ISB)	56.8 (4.7) months since mastectomy	54.1 (5.2) mean age n = 50	Mastectomy	Scapula (protraction, rotation, and tilt)	Breast cancer survivors group; Control group
Brookham et al., 2018 [4]	Humerothoracic, scapulothoracic kinematics in functional dynamic task	Vicon Motion System (Optoelectronic system) Task of shoulder ROM, ADL task (personal body care activities), and work task (reaching task with and without loads)	Recommendations of the International Society of Biomechanics (ISB)	>3 months after completing treatment	59.4 (9.7) mean age n = 50	Mastectomy, lumpectomy, axillar node dissection	Scapulothoracic (upward/downward rotation, anterior/posterior tilt, retraction/protraction) Humerothoracic angle (elevation angle; plane of elevation: flexion, abduction; external/internal rotation)	Affected side group; Unaffected side group
Maciukiewicz et al., 2022 [23]	Range of motion and strength follow breast cancer treatment	Vicon Motion System (Optoelectronic system) Arm flexion, extension, abduction, adduction, scapular abduction, and internal–external rotation	Recommendations of the International Society of Biomechanics	>1 year after surgery	53.9 (10.3) mean age n = 29	Any form of surgical procedure for breast tumor removal	Humerothoracic joint range of motion (flexion, extension, scapular abduction, abduction, internal and external rotation)	One year of treatment ending group; Between one and two years after treatment ended group
Ribeiro et al., 2019 [5]	Scapular kinematics during elevation of the arm	Flock of birds (Electromagnetic system) Elevation of the arm in the scapular plane	Recommendations of the International Society of Biomechanics	24.2 (20.5) median months since surgery	50.2 (9.8) mean age n = 42	Conserving surgery or mastectomy	Scapula (internal/external rotation, upward rotation, anterior/posterior tilt)	Control group; Surgery group
Lang et al., 2022 [7]	Breast reconstruction on kinematics during functional task	Vicon Motion System (Optoelectronic system) Maximum arm abduction and extension. Additionally, functional task (right and left repetitive reach, right and left fingertip dexterity, overhead lift, and overhead work)	Recommendations of the International Society of Biomechanics (ISB)	56.8 (42.1) mean months after surgery	54.0 (5.1) mean age n = 95	Mastectomy, reconstruction post-mastectomy, and lymph node removal	Shoulder range of motion (abduction and extension). Thoracohumeral (abduction, horizontal flexion, axial rotation). Additionally, scapular (protraction/retraction, upward/downward rotation, anterior/posterior tilt)	Control group; Mastectomy-only group; Mastectomy with reconstruction
Lopot et al., 2019 [24]	Extent of breathing movements of the thoracic and abdominal wall	Qualisys (Optoelectronic kinematic analyzer) Mild and deep breath	Global coordinate system, trajectories (X, Y, Z)	>5 years after surgery	64.5 average age n = 12	Total breast mastectomy	Thoracic and abdominal wall	Control health group; Mastectomy group
Spinelli et al., 2016 [27]	Range of motion during functional task	Liberty Polhemus (electromagnetic system) Functional task (unweighted and weighted overhead reaching, and simulated hair combing)	Recommendations of the International Society of Biomechanics	29.4 (10.8) mean months since surgery	53.8 (10.9) mean age n = 60	Lumpectomy, mastectomy, and lymph node surgery	Scapulothoracic range of motion (elevation, internal/external rotation, clavicular elevation/retraction, upward rotation, posterior til) Glenohumeral range of motion (adduction, external rotation)	Control group; Breast cancer treatment group
Rundquist et al., 2015 [20]	Lymphedema and decreased range of motion	The Motion Monitor 3D (electromagnetic system) Shoulder flexion, abduction, external and internal rotation	Local coordinate system.	72.3 (64.3) mean months from surgery	57.8 (10.1) mean age n = 30	Mastectomy	Shoulder range of motion (flexion, abduction, external and internal rotation)	Involved upper extremity; Uninvolved upper extremity
Shamley et al., 2014 [25]	Muscle activity and movement deviations	The 3 Space Fastrak (electromagnetic system) Arm elevation and depression in the scapular plane	International Shoulder Group (ISG) protocol	1143.8 (534.7) mean days after surgery	61.6 (9.1) mean age n = 176	Mastectomy and wide local excision	Scapula (internal/external rotation, upward/downward rotation, anterior/posterior tilt)	Affected side left and right; Unaffected side left and right

## Data Availability

Data from this research are not available elsewhere. Please contact the author for more information.

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
