# Peer review of "The Importance of the Kinematic Evaluation Methods of the Upper Limbs in Women with Breast Cancer Mastectomy: A Literature Review"

_healthcare, 2023, doi:10.3390/healthcare11142064_

Round 1

Reviewer 1 Report

The manuscript  is a systematic review of the technical features of kinematic methods for evaluating upper limb function in women with breast cancer treated with mastectomy. In the introduction, the authors pointed attention on the difficulties to choose a specific treatment strategy. The title of the article corresponds to its content. The article is well written and organised. The authors used several databases for literature sources, incl. PubMed, Science Direct, Clinical key, Google Scholar and Scopus. In the materials and methods, the authors provided the search strategy and clear inclusive criteria. Thus,  20 articles out of 641 were selected for a deeper analysis.  The publications were divided into a three groups depending on the methodology used to assess the kinematics of the upper limbs. In the brief discussion, the authors reported on the major changes in movement of the upper limb affected by mastectomy. The authors draw several conclusions, focusing on the importance of the movements screening in order to choose a rehabilitation program for patients.

The following improvement is recommended before the publication:

Authors should provide some deeper insight in the methodology of  kinematics evaluation in the Introduction part.

Author Response

Thank you very much for reviewing our manuscript entitled “The importance of the kinematic evaluation methods of the upper limbs in women with breast cancer mastectomy: A systematic review”. We sincerely appreciate the comments that allow us to improve the article.

In relation to the comment made, we have provided a deeper analysis of the methodology of the kinematic assessments in breast cancer survivors. This information was included in the introduction section as requested.

Note. The changes in the manuscript are marked up using the “Track Changes” function of the Microsoft Word program.

Reviewer 2 Report

The article reviews motion analysis systems used to assess upper extremity function after mastectomy surgery. The databases are well chosen, but the authors do not have enough clinical experience. Minimal movement restrictions or impairments depend on several circumstances that are not discussed:

1. The extent of mastectomy and the method of choosing the tactics of the operation.

2. It is unclear whether it is a partial (segmental) or complete mastectomy.

Breast size, whether surrounding muscular structures are involved.

Is it a complex operation together with the removal of signal lymph nodes, which expands the extent of the operation, and at the same time, the damage?

4. The patient's obesity, age, and training can have an influence.

Accordingly, the work is not fully explored.

Author Response

Thank you very much for reviewing our manuscript; we sincerely appreciate your commentaries. We have made some changes according to your suggestions to improve the quality of the document. The responses are below.

  1. We agree with your comment. The type of mastectomy and the method selected will have an effect on the kinematic movement of breast cancer survivors. Furthermore, the operation surgery technique will depend on the surgeon’s experience and the breast cancer stage. We have included some information in the results section to comply with your suggestion but we believe that this is out of the scope of the review.
  2. Some studies mention the type of surgery performed but not all of them describe this information. In some cases, the authors mention if the surgery includes a wide local excision, if it is radical mastectomy or not, if there is lymph node dissection, and so on. This information is included in the result section.
  3. Thanks for the comment, we have added the following sentence in the discussion section: “Although the treatments focus mainly on the breast tissue, they can affect the fascia, pectoral muscle tissue, ligaments, tendons, or nerves”.
  4. We agree with your comment. The body mass index will have an effect on the analysis of the kinematics of the shoulder movement. We have added the following sentence in the discussion section: “According to Lang et al., 2022 [18], the body mass index (BMI) can influence in the monitoring of the scapular movement. The presence of adipose tissue makes it difficult to locate and palpate the bony landmarks for the placement of markers. Therefore, people with overweight or obesity may present greater limitations during the kinematic evaluation”. Furthermore, it is known that rehabilitation training will help to recover breast cancer patients. This is described in the results section by Lang et al., [7] and García-Gonzalez et al., [9].

Note. The changes in the manuscript are marked up using the “Track Changes” function of the Microsoft Word program.

Reviewer 3 Report

It would be helpful to provide more specific details about the different methodologies used for kinematic evaluation, such as the specific devices or systems used within the optoelectronic and electromagnetic categories.

Clarify the criteria used for screening the title and summary of the investigations to arrive at the final selection of 20 manuscripts.

Provide examples of the different variables assessed in the studies, including specific parameters related to shoulder kinematics.

Consider expanding on the challenges posed by the heterogeneity of methodologies and variables, and discuss potential implications for comparing findings and determining patient recovery.

Clarify the intended audience of this review and specify the importance of understanding the movement of the upper limbs for the medical staff.

required editing 

Author Response

Thanks a lot for the comments, in order to improve the quality of the manuscript a substantial modification has been made. Please find below the answers to your comments.

Thanks for the comment, this information (kinematic evaluation) can be found in the results section. There, each study of the review is described in detail.

In order to clarify the inclusion criteria for selecting the articles, we have added the following sentence in the materials and methods section: “When the search was done, some of the following keywords had to appear in the title and in the abstract of the manuscripts: kinematics, breast cancer or mastectomy, motion or three-dimensional movement”. Moreover, Table 1 describes in more detail the inclusion criteria used to include articles in the review.

In section 2.4 Data collection and extraction, different variables considered in the study are mentioned. “The following variables were considered in the review: the objective of the study, the type of motion capture system (MoCS) used for the kinematic evaluation, the placement of markers and reference coordinate system considered in the studies, the time in which the kinematic assessment was performed (days, months or years before or after mastectomy), the anthropometric characteristics of the population (age, sample size, type of surgery), the region of the body evaluated (shoulder, scapula, scapulothoracic joint), the movements evaluated, and the comparison of the kinematic results (healthy subjects, contralateral arm, premeasurement, and posttest)”.

Thanks for your comments we have specified the intended audience of this review. In the abstract, we have added the following sentence “In conclusion, the interpretation of the movement of the upper limbs should be easy to understand for oncologists, physiotherapists, clinicians, and researchers”.

Thank you very much for the comment, we have expanded the discussion section in order to clarify the necessity of developing further research in order to improve the recovery process of breast cancer patients. Moreover, we have commented on the necessity of having an easy way to understand the kinematic assessment in breast cancer survivors. 

We hope that the substantial modifications made comply with the satisfaction of the doubts and comments made, thanks in advance. 

The document has been carefully reviewed to correct grammar and typo errors. 

Note. The changes in the manuscript are marked up using the “Track Changes” function of the Microsoft Word program.

Round 2

Reviewer 2 Report

no comments

Reviewer 3 Report

No any novelty 

Its very difficult to understand